# LOW-PRECISION STOCHASTIC GRADIENT LANGEVIN DYNAMICS

## ABSTRACT

Low-precision optimization is widely used to accelerate large-scale deep learning. Despite providing better uncertainty estimation and generalization, sampling methods remain mostly unexplored in this space. In this paper, we provide the first study of low-precision Stochastic Gradient Langevin Dynamics (SGLD), arguing that it is particularly suited to low-bit arithmetic due to its intrinsic ability to handle system noise. We prove the convergence of low-precision SGLD on strongly log-concave distributions, showing that with full-precision gradient accumulators, SGLD is more robust to quantization error than SGD; however, with low-precision gradient accumulators, SGLD can diverge arbitrarily far from the target distribution with small stepsizes. To remedy this issue, we develop a new quantization function that preserves the correct variance in each update step. We demonstrate that the resulting low-precision SGLD algorithm is comparable to full-precision SGLD and outperforms low-precision SGD on deep learning tasks.

## 1 INTRODUCTION

Low-precision optimization has become increasingly popular in reducing compute and memory costs of training deep neural networks (DNNs). It uses fewer bits to represent numbers in model parameters, activations and gradients and thus can drastically lower resource demands (Gupta et al., 2015; Zhou et al., 2016; De Sa et al., 2017; Li et al., 2017). Prior work has shown that using 8-bit numbers in training DNNs achieves about $4\times$ latency speed ups and memory reduction compared to 32-bit numbers on a variety of deep learning tasks (Sun et al., 2019; Yang et al., 2019; Wang et al., 2018b; Banner et al., 2018). As datasets and architectures grow rapidly, performing low-precision optimization enables training large-scale DNNs efficiently and enables many applications on different hardware and platforms.

Despite the impressive progress in low-precision optimization, low-precision sampling remains largely unexplored. However, we believe stochastic gradient Markov chain Monte Carlo (SGMCMC) methods (Welling & Teh, 2011; Chen et al., 2014; Ma et al., 2015) are particularly suited for low-precision arithmetic because of their intrinsic robustness to system noise. In particular: (1) SGMCMC explores weight space instead of converging to a single point, thus it should not require precise weights or gradients; (2) SGMCMC even adds noise to the system to encourage exploration and so is naturally more tolerant to quantization noise; (3) SGMCMC performs Bayesian model averaging during testing using an ensemble of models, which enables coarse representations of individual models to be compensated by the overall model average (Zhu et al., 2019).

SGMCMC is particularly compelling in Bayesian deep learning due to its ability to characterize complex and multimodal DNN posteriors, providing state-of-the-art generalization accuracy and calibration (BNNs) (Zhang et al., 2020; Li et al., 2016; Gan et al., 2016). Moreover, low-precision approaches are especially appealing in this setting, where at test time we must store samples from a posterior over millions of parameters, and perform multiple forward passes through the corresponding models, which incurs significant memory and computational expenses.

In this paper, we provide the first comprehensive study of low-precision Stochastic Gradient Langevin Dynamics (SGLD) (Welling & Teh, 2011). We start by analyzing the theoretical convergence of low-precision SGLD on strongly log-concave distributions, proving that *SGLD with full-precision gradient accumulators* (SGLDLP-F) converges to the target distribution within a distance that is asymptotically smaller than the distance between the SGD estimation and the optimum.

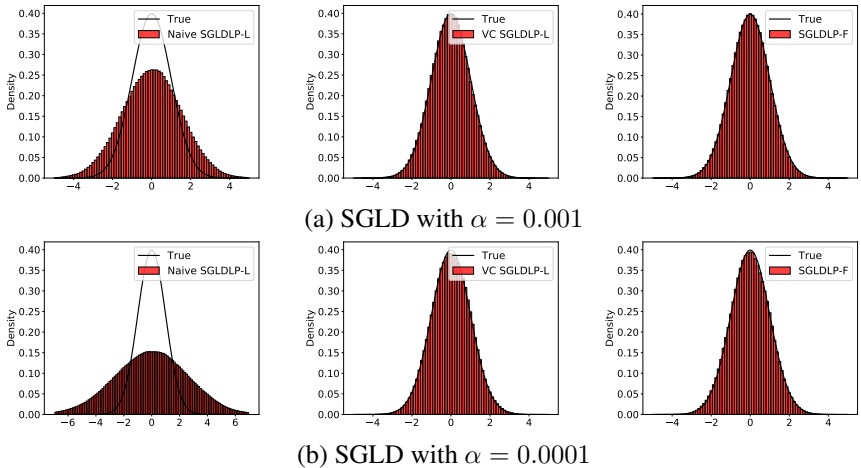

(a) SGLD with $\alpha = 0.001$

(b) SGLD with $\alpha = 0.0001$

Figure 1: Low-precision SGLD with varying stepsizes on a Gaussian distribution. Variance-corrected SGLD with low-precision gradient accumulators (VC SGLDLP-L) and SGLD with full-precision gradient accumulators (SGLDLP-F) converge to the true distribution whereas naive SGLDLP-L diverges and the divergence increases as the stepsize decreases.

Surprisingly, we find that *SGLD with low-precision gradient accumulators* (SGLDLP-L) can diverge arbitrarily far away from the target distribution with small stepsizes. We solve this issue by developing a new quantization function that preserves the correct variance in each update step. We prove that when using this new quantization function, SGLD converges to the target distribution within a bounded level of accuracy even with small stepsizes. We illustrate the behavior of different low-precision SGLD variants in Figure 1.

Empirically we demonstrate low-precision SGLD on deep learning tasks, including CIFAR-10, CIFAR-100, and ImageNet with a ResNet-18, and the IMDB dataset with a LSTM. We show that our variance-corrected quantizer significantly improves the performance of low-precision SGLD. Moreover, the improvement of SGLD over SGD is larger in low-precision than in full-precision, demonstrating the promise of low-precision stochastic sampling.

## 2 RELATED WORK

Most work on accelerating SGLD training has focused on distributed learning with synchronous or asynchronous communication (Ahn et al., 2014; Chen et al., 2016; Li et al., 2019). Another direction to reduce training costs is to shorten training time by accelerating the convergence using either variance reduction techniques (Dubey et al., 2016; Baker et al., 2019) or a cyclical learning rate schedule (Zhang et al., 2020). To speed up testing, distillation techniques are often used to save both compute and memory by transferring the knowledge of an ensemble of models to a single model (Korattikara et al., 2015; Wang et al., 2018a).

Low-precision computation has become one of the most common approaches to reduce latency and memory consumption in deep learning and is widely supported on new emerging chips including CPUs, GPUs and TPUs (Micikevicius et al., 2017; Krishnamoorthi, 2018; Esser et al., 2019). Two main directions to improve low-precision training include developing new number formats (Sun et al., 2019; 2020) and studying mixed precision schemes (Courbariaux et al., 2015; Zhou et al., 2016; Banner et al., 2018). Recently, one line of work studies applying Bayesian framework to learn a deterministic quantized neural network (Soudry et al., 2014; Cheng et al., 2015; Achterhold et al., 2018; van Baalen et al., 2020; Meng et al., 2020).

Despite impressive progress of low-precision deep learning, few work takes advantage of it for Bayesian neural networks (BNNs). Su et al. (2019) proposes a method to train binarized variational BNNs and Cai et al. (2018) develops an efficient hardware for training low-precision variational BNNs. The only work on low-precision MCMC known to us is Ferianc et al. (2021), which directly applies *post-training* quantization in optimization (Jacob et al., 2018) to quantize BNNs trained by

Stochastic Gradient Hamiltonian Monte Carlo (Chen et al., 2014). In contrast, we study training low-precision models by SGLD from scratch, which can reduce both training and testing costs.

## 3 PRELIMINARIES

### 3.1 STOCHASTIC GRADIENT LANGEVIN DYNAMICS

In the Bayesian setting, given some dataset $\mathcal{D}$, a model with parameters $\theta$, and a prior $p(\theta)$, we are interested in sampling from the posterior $p(\theta|\mathcal{D}) \propto \exp(-U(\theta))$, where the energy function is

$$U(\theta) = -\sum_{x \in \mathcal{D}} \log p(x|\theta) - \log p(\theta).$$

When the dataset is large, the cost of computing a sum over the entire dataset is expensive. Stochastic Gradient Langevin Dynamics (SGLD) reduces the cost by using a stochastic gradient estimation $\nabla \tilde{U}$, an unbiased estimator of $\nabla U$ usually based on a subset of the dataset $\mathcal{D}$. Specifically, SGLD updates the parameter $\theta$ in the $(k+1)$-th step following the rule

$$\theta_{k+1} = \theta_k - \alpha \nabla \tilde{U}(\theta_k) + \sqrt{2\alpha} \xi_{k+1},$$

where $\alpha$ is the stepsize and $\xi$ is a standard Gaussian noise. Compared to the SGD update, the only difference is the additional Gaussian noise. This close connection makes SGLD convenient to implement on existing deep learning tasks for which SGD is the typical learning algorithm.

### 3.2 LOW-PRECISION TRAINING

We study training a low-precision model by SGLD from scratch, to reduce both training and testing costs. We use *fixed point* in our theoretical analysis and empirical demonstration, which is a common number type for reducing DNNs costs (Gupta et al., 2015; Lin et al., 2016; Li et al., 2017; Yang et al., 2019). We additionally use block floating point, another common number representation, for empirical evaluations on deep learning tasks (Song et al., 2018).

In order to train a network using low-precision numbers, we need a quantization function $Q$ to convert a real-valued number into a rounded number. Such functions include *deterministic rounding* and *stochastic rounding*. Suppose that we use $W$ bits to represent numbers with $F$ of those $W$ bits to represent the fractional part. Then the *quantization gap* $\Delta = 2^{-F}$ is the distance between consecutive representable numbers. The lower and upper bounds of the representable numbers are $l = -2^{W-F-1}$ and $u = 2^{W-F-1} - 2^{-F}$, respectively. The deterministic rounding quantizes a number to its nearest representable neighbour:

$$Q^d(\theta) = \text{sign}(\theta) \cdot \text{clip}\left(\Delta \left\lfloor \frac{|\theta|}{\Delta} + \frac{1}{2} \right\rfloor, l, u\right),$$

where $\text{clip}(x, l, u) = \max(\min(x, u), l)$. Stochastic rounding quantizes a number with probability:

$$Q^s(\theta) = \begin{cases} \text{clip}\left(\Delta \left\lfloor \frac{\theta}{\Delta} \right\rfloor, l, u\right), & \text{w.p. } \left\lceil \frac{\theta}{\Delta} \right\rceil - \frac{\theta}{\Delta} \\ \text{clip}\left(\Delta \left\lceil \frac{\theta}{\Delta} \right\rceil, l, u\right), & \text{w.p. } 1 - \left(\left\lceil \frac{\theta}{\Delta} \right\rceil - \frac{\theta}{\Delta}\right). \end{cases}$$

$Q^s$ is unbiased, which means $\mathbf{E}[Q^s(\theta)] = \theta$. It is more favorable than $Q^d$ in practice since it can preserve gradient information when the gradient update is smaller than the quantization gap (Gupta et al., 2015; Courbariaux et al., 2016). In what follows, we use $Q_w$ and $\Delta_w$ to denote the weights' quantizer and quantization gap, $Q_g$ and $\Delta_g$ to denote gradients' quantizer and quantization gap. Please note that $Q_g$ means quantizing the error and gradient in each layer in backward propagation (Wu et al., 2018; Yang et al., 2019).

Besides the quantization function, we also need to decide what and where to quantize. There are two common choices depending on whether we store an additional copy of full-precision weights. *Full-precision gradient accumulators* use a full-precision weight buffer to accumulate gradient updates and only quantize weights before computing gradients. SGD with full-precision gradient accumulators (SGDLP-F) updates the weights as

$$\theta_{k+1} = \theta_k - \alpha Q_g\left(\nabla \tilde{U}(Q_w(\theta_k))\right).$$

Gradient accumulators are frequently updated during training, therefore it will be ideal to also represent it in low-precision to further reduce the costs. The update of SGD with *low-precision gradient accumulators* (SGDLP-L) is

$$\theta_{k+1} = Q_w \left( \theta_k - \alpha Q_g \left( \nabla \tilde{U}(\theta_k) \right) \right).$$

Both full- and low-precision gradient accumulators have been widely used in low-precision training. Low-precision gradient accumulators are cheaper and full-precision gradient accumulators generally have better performance because of more precisely accumulating gradient updates (Courbariaux et al., 2015; Li et al., 2017).

## 4 LOW-PRECISION SGLD

In this section, we first study the convergence bound of low-precision SGLD with full-precision gradient accumulators (SGLDLP-F) on strongly log-concave distributions and show that SGLDLP-F converges to the target distribution within a distance that is asymptotically smaller than that between the SGD estimation and the optimum. Next we analyze low-precision SGLD with low-precision gradient accumulators (SGLDLP-L) under the same setup and prove that SGLDLP-L can diverge arbitrarily far away from the target distribution with a small stepsize, which is typically required by SGLD to reduce asymptotic bias. Finally, we solve this problem by developing a variance-corrected quantization function and further prove that our SGLDLP-L converges with small stepsizes.

### 4.1 FULL-PRECISION GRADIENT ACCUMULATORS

Due to the similarity between SGLD and SGD, applying low-precision to SGLD is straightforward. Similar to SGDLP-F, we can do low-precision SGLD with full-precision gradient accumulators (SGLDLP-F) as the following.

$$\theta_{k+1} = \theta_k - \alpha Q_g \left( \nabla \tilde{U}(Q_w(\theta_k)) \right) + \sqrt{2\alpha}\xi_{k+1}. \tag{1}$$

We now prove that SGLDLP-F will converge to the target distribution given small stepsizes and large number of iterations.

Our convergence analysis of low-precision SGLD is based on (Dalalyan & Karagulyan, 2019), where the target distribution is assumed smooth and strongly log-concave. We additionally assume the energy function has Lipschitz Hessian following recent work in low-precision optimization (Yang et al., 2019). Specifically, the energy function $U$ satisfies the following,

$$\begin{cases} U(\theta) - U(\theta') - \nabla U(\theta')^{\mathsf{T}}(\theta - \theta') \geq (m/2) \|\theta - \theta'\|_2^2, \\ \|\nabla U(\theta) - \nabla U(\theta')\|_2 \leq M \|\theta - \theta'\|_2, \qquad \forall \theta, \theta' \in \mathbb{R}^d \\ \|\nabla^2 U(\theta) - \nabla^2 U(\theta')\|_2 \leq \Psi \|\theta - \theta'\|_2, \end{cases}$$

for some positive constants $m$, $M$ and $\Psi$. We further assume the variance of the stochastic gradient is bounded by $\mathbf{E}[\|\nabla \tilde{U}(\theta) - \nabla U(\theta)\|_2^2] \leq \kappa^2$ for some constant $\kappa$. For simplicity, we consider SGLD with a constant stepsize $\alpha$. We measure the convergence of SGLD in terms of 2-Wasserstein distance. We use stochastic rounding for both weight and gradient quantizers as it is generally better than deterministic rounding and has also been used in previous low-precision theoretical analysis (Li et al., 2017; Yang et al., 2019).

**Theorem 1.** *We run SGLDLP-F under the above assumptions and with a constant stepsize $\alpha \leq 2/(m+M)$. Let $\pi$ be the target distribution, $\mu_K$ be the distribution obtained after $K$ iterations and $\mu_0$ be the initial distribution, then*

$$W_2(\mu_K, \pi) \leq (1 - \alpha m)^K W_2(\mu_0, \pi) + 1.65(M/m)(\alpha d)^{1/2} + \min\left( \frac{\Psi \Delta_w^2 d}{4m}, \frac{M \Delta_w \sqrt{d}}{2m} \right)$$

$$+ \sqrt{\frac{(\Delta_g^2 + M^2 \Delta_w^2)\alpha d + 4\alpha \kappa^2}{4m}}.$$

There are three main takeways of this theorem: First, it shows that SGLDLP-F converges to the accuracy floor $\min\left(\frac{\Psi\Delta_w^2 d}{4m}, \frac{M\Delta_w\sqrt{d}}{2m}\right)$ which depends on the quantization gap $\Delta_w$ given large enough $K$ and small enough $\alpha$. Second, our convergence bound is $O(\Delta_w^2)$ which is better than the distance between the SGD estimation and the optimum $O(\Delta_w)$ (Yang et al., 2019) (see Appendix D for details). Third, if we further assume the energy function is quadratic, that is $\Psi = 0$, then SGLDLP-F converges to the target distribution asymptotically. This is similar to SGDLP-F on quadratic function which converges to the optimum asymptotically (Li et al., 2017).

## 4.2 LOW-PRECISION GRADIENT ACCUMULATORS

To further reduce the costs, we apply low-precision gradient accumulators to SGLD which we denote SGLDLP-L. Mimicking the update of SGDLP-L, the update rule of SGLDLP-L is

$$\theta_{k+1} = Q_w\left(\theta_k - \alpha Q_g\left(\nabla\tilde{U}(\theta_k)\right) + \sqrt{2\alpha}\xi_{k+1}\right). \tag{2}$$

Surprisingly, while we can prove a convergence result for SGLDLP-L, our theory suggests that it can diverge arbitrarily from the target distribution with small stepsizes.

**Theorem 2.** *We run SGLDLP-L under the same assumptions as in Theorem 1. Then*

$$W_2(\mu_K, \pi) \leq (1 - \alpha m)^K W_2(\mu_0, \pi) + 1.65(M/m)(\alpha d)^{1/2} + \min\left(\frac{\Psi\Delta_w^2 d}{4m}, \frac{M\Delta_w\sqrt{d}}{2m}\right)$$

$$+ \sqrt{\frac{(\alpha\Delta_g^2 + \alpha^{-1}\Delta_w^2)d + 4\alpha\kappa^2}{4m}} + \left((1 - \alpha m)^K + 1\right)\frac{\Delta_w\sqrt{d}}{2}.$$

This theorem suggests that as the stepsize $\alpha$ decreases, $W_2$ distance between the stainary distribution of SGLDLP-L and the target distribution increases. When $\alpha$ decreases, either our bound becomes loose or SGLDLP-L diverges from the target distribution. We empirically test SGLDLP-L on a standard Gaussian distribution in Figure 1, showing that the reason is the later. We use 8-bit fixed point and assign 2 of them to represent the integer part. We can see that SGLDLP-F always converges to the target distribution with different stepsizes whereas SGLDLP-L diverges from the target distribution and the divergence increases when the stepsize decreases.

One may choose a stepsize that minimizes the above $W_2$ distance to avoid divergence, however, getting this optimal stepsize is generally difficult as the constants are unknown in practice. Moreover, enabling a small stepsize in SGLD is desirable, since SGLD needs a small stepsize to reduce asymptotic bias of the posterior approximation (Welling & Teh, 2011).

## 4.3 VARIANCE-CORRECTED QUANTIZATION

To approach correcting the problems with naïve SGLDLP-L, we first need to identify the source of the issue. We start by showing that the reason is the *variance* of each dimension of $\theta_{k+1}$ becomes larger due to using stochastic rounding as the low-precision gradient accumulators. Specifically, given the stochastic gradient $\nabla\tilde{U}$, the update of full-precision SGLD can be viewed as sampling from a Gaussian distribution for each dimension $i$

$$\theta_{k+1,i} \sim \mathcal{N}\left(\theta_{k,i} - \alpha\nabla\tilde{U}(\theta_k)_i, 2\alpha\right), \text{for } i = 1, \cdots, d.$$

Using stochastic rounding as the weight quantizer and the gradient quantizer in SGLDLP-L gives us

$$\mathbf{E}\left[\theta_{k+1,i}\right] = \mathbf{E}\left[Q^s\left(\theta_{k,i} - \alpha Q^s\left(\nabla\tilde{U}(\theta_k)\right)_i + \sqrt{2\alpha}\xi_{k+1,i}\right)\right] = \theta_{k,i} - \alpha\nabla\tilde{U}(\theta_k)_i$$

which keeps the same mean of $\theta_{k+1}$ as in full-precision. However the variance of $\theta_{k+1,i}$ is larger than needed. If we ignore the variance from $Q_g$ and the stochastic gradient, since they are present and have been shown to work well in SGLDLP-F, the variance of $\theta_{k+1,i}$ is

$$\text{Var}\left[\theta_{k+1,i}\right] = \mathbf{E}\left[\text{Var}\left[Q^s\left(\theta_{k,i} - \alpha\nabla U(\theta_k)_i + \sqrt{2\alpha}\xi_{k+1,i}\right)\Big|\xi_{k+1,i}\right]\right]$$

$$+ \text{Var}\left[\mathbf{E}\left[Q^s\left(\theta_{k,i} - \alpha\nabla U(\theta_k)_i + \sqrt{2\alpha}\xi_{k+1,i}\right)\Big|\xi_{k+1,i}\right]\right]$$

$$= \frac{\Delta_w^2}{4}\chi_{k+1,i} + 2\alpha.$$

---

**Algorithm 1** Variance-Corrected Low-Precision SGLD (VC SGLDLP-L).

---

**given:** Stepsize $\alpha$, number of training iterations $K$, gradient quantizer $Q_g$, deterministic rounding $Q^d$, stochastic rounding $Q^s$ and quantization gap of weights $\Delta_w$.

**for** $k = 1 : K$ **do**

    **update** $\theta_{k+1} \leftarrow Q^{vc}\left(\theta_k - \alpha Q_g\left(\nabla\tilde{U}(\theta_k)\right), 2\alpha, \Delta_w, Q^d, Q^s\right)$

**end for**

**output:** samples $\{\theta_k\}$

---

where $\chi_{k+1,i} \in [0, 1]$ depends on the distance of $\theta_{k+1,i}$ to the discrete grid. Empirically, from Figure 1, we can see that naïve SGLDLP-L gives the correct mean estimate but the wrong variance estimate, while both are estimated correctly in SGLD which adds the proper amount of noise in each update. This validates our intuition that using stochastic rounding in low-precision gradient accumulators adds more variance than needed leading to an inaccurate variance estimate.

To address this issue, we propose a new quantization function $Q^{vc}$ in Algorithm 2. $Q^{vc}$ always guarantees the correct mean, $\mathbf{E}[\theta_{k+1,i}] = \theta_{k,i} - \alpha\nabla\tilde{U}(\theta_k)_i$, and guarantees the correct variance $\text{Var}[\theta_{k+1,i}] = 2\alpha$ most of the time except when $v < v_s$. However that case rarely happens in practice, because the stepsize has to be extremely small. The main idea of $Q^{vc}$ is to directly sample from the discrete weight space instead of quantizing a real-valued Gaussian sample. It does so considering two cases: when the Gaussian noise is larger than the largest possible stochastic rounding variance $\Delta^2/4$, $Q^{vc}$ first adds a portion of the Gaussian noise and uses a sample from the weight grid to make up the remaining; in the other situation, $Q^{vc}$ directly samples from the weight grid to achieve the target variance. Although $Q^{vc}$ does not provide a sample following a *Gaussian* distribution (as it is discrete), we show that this does not affect the performance in both theory and practice.

We now prove that SGLDLP-L using $Q^{vc}$, denoting VC SGLDLP-L, converges to the target distribution up to a certain accuracy level with small stepsizes.

**Theorem 3.** *We run VC SGLDLP-L as in Algorithm 1. Besides the same assumptions in Theorem 1, we further assume the gradient is bounded* $\mathbf{E}\left[\left\|Q_g(\nabla\tilde{U}(\theta_k))\right\|_1\right] \leq G$. *Let* $v_0 = \Delta_w^2/4$. *Then*

$$W_2(\mu_K, \pi) \leq (1 - \alpha m)^K W_2(\mu_0, \pi) + 1.65(M/m)(\alpha d)^{1/2} + \min\left(\frac{\Psi A}{m}, \frac{M\sqrt{A}}{m}\right)$$

$$+ \sqrt{\frac{\alpha\Delta_g^2 d + 4\alpha\kappa^2}{4m} + \frac{A}{\alpha m}} + \left((1 - \alpha m)^K + 1\right)\sqrt{A}.$$

*where* $A = \begin{cases} 5v_0 d, & \text{if } 2\alpha > v_0 \\ \max\left(2\Delta_w\alpha G, 4\alpha d\right), & \text{otherwise} \end{cases}$

This theorem shows that when the stepsize $\alpha \to 0$, VC SGLDLP-L converges to the target distribution up to an error instead of diverging. Our bound is $O(\sqrt{\Delta_w})$ which is equivalent to the distance between SGD with low-precision gradient accumulators and the optimum (Li et al., 2017; Yang et al., 2019). However, we show empirically that VC SGLDLP-L has better dependency on the quantization gap than SGD. We leave the improvement of the theoretical bound for future work.

We empirically demonstrate VC SGLDLP-L on the standard Gaussian distribution under the same setting as in the previous section in Figure 1. Regardless of the stepsize, VC SGLDLP-L converges to the target distribution and approximates the target distribution as well as SGLDLP-F, showing that preserving the correct variance is the key to ensure correct convergence.

## 5 EXPERIMENTS

We demonstrate low-precision SGLD with full-precision gradient accumulators (SGLDLP-F) and with variance-corrected low-precision gradient accumulators (VC SGLDLP-L) on a logistic regression and multilayer perceptron on MNIST dataset (Section 5.1), ResNet-18 on CIFAR datasets and LSTM on IMDB dataset (Section 5.2), and ResNet-18 on Imagenet dataset (Section 5.3). We use qtorch (Zhang et al., 2019) to simulate low-precision training on these experiments.

---

**Algorithm 2** Variance-Corrected Quantization Function $Q^{vc}$.

---

**input**: $(\mu, v, \Delta, Q^d, Q^s)$      // $Q^{vc}$ returns a variable with mean $\mu$ and variance $v$
$v_0 \leftarrow \Delta^2/4$      // $\Delta^2/4$ is the largest possible variance that stochastic rounding can cause
**if** $v > v_0$ **then**
     $x \leftarrow \mu + \sqrt{v - v_0}\xi$, where $\xi \sim \mathcal{N}(0, I_d)$
     $r \leftarrow x - Q^d(x)$
     **for all** $i$ **do**
$$c_i \leftarrow \begin{cases} \Delta, & w.p. \frac{v_0 + r_i^2 + |r_i|\Delta}{2\Delta^2} \\ -\Delta, & w.p. \frac{v_0 + r_i^2 - |r_i|\Delta}{2\Delta^2} \\ 0, & \text{otherwise} \end{cases}$$
         // When $v > v_0$, we add a portion of Gaussian noise and
         // sample from the weight grid to make up the remaining
     **end for**
     $\theta \leftarrow Q^d(x) + \text{sign}(r) \odot c$
**else**
     $r \leftarrow \mu - Q^s(\mu)$
     **for all** $i$ **do**
         $v_s \leftarrow \left(1 - \frac{|r_i|}{\Delta}\right) \cdot r_i^2 + \frac{|r_i|}{\Delta} \cdot (-r_i + \text{sign}(r_i)\Delta)^2$
         **if** $v > v_s$ **then**
$$c_i \leftarrow \begin{cases} \Delta, & w.p. \frac{v - v_s}{2\Delta^2} \\ -\Delta, & w.p. \frac{v - v_s}{2\Delta^2} \\ 0, & \text{otherwise} \end{cases}$$
             // When $v \le v_0$, we sample from the weight grid
             // to achieve the target variance
         $\theta_i \leftarrow Q^s(\mu)_i + c_i$
         **else**
             $\theta_i \leftarrow Q^s(\mu)_i$
         **end if**
     **end for**
**end if**
clip $\theta$ if outside representable range
**return** $\theta$

---

## 5.1 Logistic Regression and Multilayer Perceptron

We first empirically verify our theorems on a logistic regression on MNIST dataset. We use $\mathcal{N}(0, 1/6)$ as the prior distribution following (Yang et al., 2019) and the resulting posterior distribution satisfies the assumptions in Section 4. We use fixed point numbers with 2 integer bits and vary the number of fractional bits which is corresponding to varying the quantization gap $\Delta$.

From Figure 2a, SGLDLP-F is more robust to the decay of bits than SGDLP-F since SGLDLP-F outperforms SGDLP-F on all number of bits and recovers the full-precision result with 6 bits whereas SGDLP-F needs 10 bits. This verifies Theorem 1 that SGLDLP-F converges to the target distribution up to an error and is more robust to the quantiazation gap than SGDLP-F. With low-precision gradient accumulators, we can see that VC SGLDLP-L is significantly better than naïve SGLDLP-L, indicating the effectiveness of using the variance-corrected quantization function for quantizing gradient accumulators. These results verify Theorem 2 and Theorem 3. Besides, VC SGLDLP-L outperforms SGDLP on all number of bits and even outperforms SGDLP-F when using 2 fractional bits. VC SGLDLP-L matches full-precision SGLD results with 6 bits whereas SGDLP-L needs 8 or 10 bits. These observations demonstrate that SGLD is still more favorable than SGD when using low-precision gradient accumulators.

To test whether these results apply to non-log-concave distributions, we replace the logistic regression model by a two-layer multilayer perceptron (MLP). The MLP has 100 hidden units and RELU nonlinearilities. From Figure 2b, we observe similar results as on the logistic regression, suggesting that empirically our analysis still stands on non-log-concave distributions. We also provide negative log-likelihood (NLL) comparisons in terms of number of bits on the logistic regression and MLP in Figure 3 in Appendix.

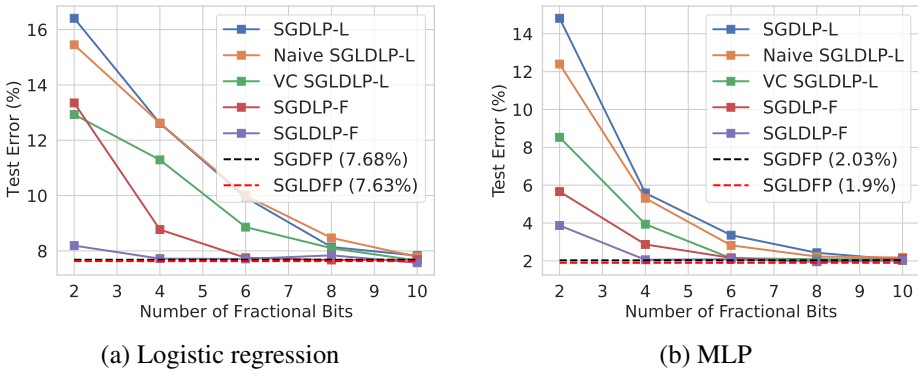

(a) Logistic regression  (b) MLP

Figure 2: Test errors on MNIST dataset in terms of different precision.

## 5.2 RESNET AND LSTM

We consider image and sentiment classification tasks: CIFAR datasets (Krizhevsky et al., 2009) on ResNet-18 (He et al., 2016), and IMDB dataset (Maas et al., 2011) on LSTM (Hochreiter & Schmidhuber, 1997). We use 8-bit number representation since it becomes increasingly popular in training deep models and is powered by new generation of chips (Sun et al., 2019; Banner et al., 2018; Wang et al., 2018b). We report test errors averaged over 3 runs with the standard error in Table 1.

**Fixed Point** We use 8-bit fixed point for weights and gradients but full-precision for activations since we find low-precision activations significantly harm the performance. Similar to the results in previous sections, SGLDLP-F is better than SGDLP-F and VC SGLDLP-L significantly outperforms naïve SGLDLP-L and SGDLP-L across datasets and architectures. Notably, the improvement of SGLD over SGD becomes larger when using more low-precision arithmetic. For example, on CIFAR-100, VC SGLDLP-L outperforms SGDLP-L by 2.24%, SGLDLP-F outperforms SGDLP-F by 0.54% and SGLD outperforms SGDLP by 0.06%. This demonstrates that SGLD is particularly compatible with low-precision deep learning because of its natural ability to handle system noise.

**Block Floating Point** We also consider block floating point (BFP) which is another common number type and is often preferred over fixed point on deep models due to less quantization error caused by overflow and underflow (Song et al., 2018). Following the block design in Yang et al. (2019), we use *small-block* for ResNet and *big-block* for LSTM. The $Q^{vc}$ function naturally generalizes to BFP and only needs a small modification (see Appendix E for the algorithm of $Q^{vc}$ with BFP). By using BFP, the results of all low-precision methods improve over fixed point. SGLDLP-F can match the performance of SGLDFP with all numbers quantized to 8-bit except gradient accumulators. VC SGLDLP-L still outperforms naive SGLDLP-L indicating the effectiveness of $Q^{vc}$ with BFP. Again, SGLDFP-F and VC SGLDLP-L outperform their SGD counterparts on all tasks, suggesting the general applicability of low-precision SGLD with different number types.

**Cyclical SGLD** We further apply low-precision to a recent variant of SGLD, *cSGLD*, which utilizes a cyclical learning rate schedule to speed up convergence (Zhang et al., 2020). We observe that the results of cSGLD-F are very close to those of cSGLDFP, and VC cSGLDLP-L can match or even outperforms full-precision SGD with all numbers quantized to 8 bits! These results indicate that diverse samples from different modes can countereffect the quantization error by providing complementary predictions.

## 5.3 IMAGENET

Finally, we test low-precision SGLD on a large scale image classification dataset, ImageNet, with ResNet-18. We train SGD for 90 epochs and train SGLD for 10 epochs using the trained SGD model as the initialization. The improvement of SGLD over SGD is larger in low-precision (0.76% top-1

Table 1: Test errors (%) on CIFAR-10, CIFAR-100 with ResNet-18 and IMDB with LSTM.

| | CIFAR-10 | CIFAR-100 | IMDB |
|---|---|---|---|
| **32-BIT FLOATING POINT** | | | |
| SGLDFP | $4.65_{\pm 0.06}$ | $22.58_{\pm 0.18}$ | $13.43_{\pm 0.21}$ |
| SGDFP | $4.71_{\pm 0.02}$ | $22.64_{\pm 0.13}$ | $13.88_{\pm 0.29}$ |
| cSGLDFP | $4.54_{\pm 0.05}$ | $21.63_{\pm 0.04}$ | $13.25_{\pm 0.18}$ |
| **8-BIT FIXED POINT** | | | |
| NAÏVE SGLDLP-L | $7.82_{\pm 0.13}$ | $27.25_{\pm 0.13}$ | $16.63_{\pm 0.28}$ |
| VC SGLDLP-L | $7.13_{\pm 0.01}$ | $26.62_{\pm 0.16}$ | $15.38_{\pm 0.27}$ |
| SGLDLP-F | $5.12_{\pm 0.06}$ | $23.30_{\pm 0.09}$ | $15.40_{\pm 0.36}$ |
| SGDLP-L | $8.53_{\pm 0.08}$ | $28.86_{\pm 0.10}$ | $19.28_{\pm 0.63}$ |
| SGDLP-F | $5.20_{\pm 0.14}$ | $23.84_{\pm 0.12}$ | $15.74_{\pm 0.79}$ |
| **8-BIT BLOCK FLOATING POINT** | | | |
| NAÏVE SGLDLP-L | $5.85_{\pm 0.04}$ | $26.38_{\pm 0.13}$ | $14.64_{\pm 0.08}$ |
| VC SGLDLP-L | $5.51_{\pm 0.01}$ | $25.22_{\pm 0.18}$ | $13.99_{\pm 0.24}$ |
| SGLDLP-F | $4.58_{\pm 0.07}$ | $22.59_{\pm 0.18}$ | $14.05_{\pm 0.33}$ |
| SGDLP-L | $5.86_{\pm 0.18}$ | $26.19_{\pm 0.11}$ | $16.06_{\pm 1.81}$ |
| SGDLP-F | $4.75_{\pm 0.05}$ | $22.9_{\pm 0.13}$ | $14.28_{\pm 0.17}$ |
| VC cSGLDLP-L | $4.97_{\pm 0.10}$ | $22.61_{\pm 0.12}$ | $13.09_{\pm 0.27}$ |
| cSGLD-F | $4.32_{\pm 0.07}$ | $21.50_{\pm 0.14}$ | $13.13_{\pm 0.37}$ |

Table 2: Test errors (%) on ImageNet with ResNet-18.

| | TOP-1 | TOP-5 |
|---|---|---|
| **32-BIT FLOATING POINT** | | |
| SGLD | 30.39 | 10.76 |
| SGD | 30.56 | 10.97 |
| **8-BIT BLOCK FLOATING POINT** | | |
| VC SGLDLP-L | 31.47 | 11.77 |
| SGDLP-L | 32.23 | 12.09 |

error) than in full-precision (0.17% top-1 error), showing the advantages of low-precision SGLD on large-scale deep learning tasks.

# 6 CONCLUSION

We provide the first comprehensive study of low-precision SGLD. In theory, we prove that with full-precision gradient accumulators, SGLD can converge to the target distribution within a distance that is asymptotically smaller than the distance between the SGD estimation and the optimum; with low-precision gradient accumulators, SGLD can diverge arbitrarily far away from the target distribution with small stepsizes. We find that the issue is caused by the wrong variance in each update and thus develop a new varaince-corrected quantization function that preserves the correct variance. We prove that SGLD with this quantization function converges to the target distribution up to a certain level depending on the quantization gap. In practice, we verify our theoretical results and demonstrate SGLDLP-F and VC SGLDLP-L are comparable to full-precision SGLD with 8-bit on image and sentiment classification tasks.

In the future, it will be interesting to extend low-precision computation to other stochastic MCMC methods and improve theoretical bounds to better reflect empirical performance. We hope this work shed lights on accelerating MCMC methods for Bayesian deep learning.

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

# A    PROOF OF THEOREM 1

Our proof follows Theorem 4 in Dalalyan & Karagulyan (2019), which provides a convergence bound of Langevin dynamics with noisy gradient. We state the result of Theorem 4 in Dalalyan & Karagulyan (2019) below.

We consider Langevin dynamics whose update rule is

$$\theta_{k+1} = \theta_k - \alpha \left( \nabla U(\theta_k) + \zeta_k \right) + \sqrt{2\alpha} \xi_{k+1}.$$

The noise in the gradient $\zeta_k$ has the following three assumptions:

$$\mathbf{E}\left[ \|\mathbf{E}\left[\zeta_k | \theta_k\right]\|_2^2 \right] \leq \delta^2 d, \quad \mathbf{E}\left[ \|\zeta_k - \mathbf{E}\left[\zeta_k|\theta_k\right]\|_2^2 \right] \leq \sigma^2 d, \quad \xi_{k+1} \text{ is independent of } (\zeta_0, \cdots, \zeta_k)$$

where $\delta > 0$ and $\sigma > 0$ are some constants. Under the same assumptions in Section 4, we have the convergence bound for the above Langevin dynamics.

**Theorem 4.** *We run the above Langevin dynamics with $\alpha \leq 2/(m + M)$. Let $\pi$ be the target distribution, $\mu_K$ be the distribution obtained after $K$ iterations and $\mu_0$ be the initial distribution. Then*

$$W_2(\mu_K, \pi) \leq (1 - \alpha m)^K W_2(\mu_0, \pi) + 1.65(M/m)(\alpha d)^{1/2} + \frac{\delta\sqrt{d}}{m} + \frac{\sigma^2(\alpha d)^{1/2}}{1.65M + \sigma\sqrt{m}}.$$

*Proof.* We write the SGLDLP-F update as the following

$$\begin{aligned}
\theta_{k+1} &= \theta_k - \alpha Q_g(\nabla \tilde{U}(Q_w(\theta_k))) + \sqrt{2\alpha} \xi_{k+1} \\
&= \theta_k - \alpha \left( \nabla U(\theta_k) + \zeta_k \right) + \sqrt{2\alpha} \xi_{k+1}
\end{aligned}$$

where

$$\begin{aligned}
\zeta_k &= Q_g(\nabla \tilde{U}(Q_w(\theta_k))) - \nabla U(\theta_k) \\
&= Q_g(\nabla \tilde{U}(Q_w(\theta_k))) - \nabla \tilde{U}(Q_w(\theta_k)) \\
&\quad + \nabla \tilde{U}(Q_w(\theta_k)) - \nabla U(Q_w(\theta_k)) + \nabla U(Q_w(\theta_k)) - \nabla U(\theta_k).
\end{aligned}$$

Since $\mathbf{E}[\nabla \tilde{U}(x)] = \nabla U(x)$ and $\mathbf{E}[Q(x)] = x$, we have

$$\begin{aligned}
\mathbf{E}[\zeta_k] &= \mathbf{E}\left[ Q_g(\nabla \tilde{U}(Q_w(\theta_k))) - \nabla \tilde{U}(Q_w(\theta_k)) \right] \\
&\quad + \mathbf{E}\left[ \nabla \tilde{U}(Q_w(\theta_k)) - \nabla U(Q_w(\theta_k)) \right] + \mathbf{E}\left[ \nabla U(Q_w(\theta_k)) - \nabla U(\theta_k) \right] \\
&= \mathbf{E}\left[ \nabla U(Q_w(\theta_k)) - \nabla U(\theta_k) \right].
\end{aligned}$$

By the assumption, we know that

$$\begin{aligned}
\mathbf{E}\left[ \|\nabla U(Q_w(\theta_k)) - \nabla U(\theta_k)\|_2^2 \right] &\leq M^2 \mathbf{E}\left[ \|Q_w(\theta_k) - \theta_k\|_2^2 \right] \\
&\leq M^2 \cdot \frac{\Delta_w^2 d}{4}.
\end{aligned}$$

It follows that

$$\begin{aligned}
\|\mathbf{E}[\zeta_k]\|_2^2 &= \|\mathbf{E}\left[ \nabla U(Q_w(\theta_k)) - \nabla U(\theta_k) \right]\|_2^2 \\
&\leq \mathbf{E}\left[ \|\nabla U(Q_w(\theta_k)) - \nabla U(\theta_k)\|_2^2 \right] \\
&\leq M^2 \cdot \frac{\Delta_w^2 d}{4}.
\end{aligned}$$

Let $f : \mathbb{R} \to \mathbb{R}^d$ denote the function

$$f(a) = \nabla U(\theta_k + a(Q_w(\theta_k) - \theta_k)).$$

By the mean value theorem, there will exist an $a \in [0, 1]$ (a function of the weight quantization randomness) such that

$$f(1) - f(0) = f'(a).$$

So,

$$
\begin{aligned}
\mathbf{E}[\zeta_k] &= \mathbf{E}\left[\nabla U(Q_w(\theta_k)) - \nabla U(\theta_k)\right] \\
&= \mathbf{E}\left[\nabla^2 U(\theta_k + a(Q_w(\theta_k) - \theta_k))(Q_w(\theta_k) - \theta_k)\right] \\
&= \mathbf{E}\left[\nabla^2 U(\theta_k)(Q_w(\theta_k) - \theta_k)\right] \\
&\quad + \mathbf{E}\left[\left(\nabla^2 U(\theta_k + a(Q_w(\theta_k) - \theta_k)) - \nabla^2 U(\theta_k)\right)(Q_w(\theta_k) - \theta_k)\right] \\
&= \mathbf{E}\left[\left(\nabla^2 U(\theta_k + a(Q_w(\theta_k) - \theta_k)) - \nabla^2 U(\theta_k)\right)(Q_w(\theta_k) - \theta_k)\right].
\end{aligned}
$$

Now, by the assumption $\|\nabla^2 U(x) - \nabla^2 U(y)\|_2 \le \Psi \|x - y\|$, we get

$$
\begin{aligned}
\|\mathbf{E}[\zeta_k]\| &= \left\|\mathbf{E}\left[\left(\nabla^2 U(\theta_k + a(Q_w(\theta_k) - \theta_k)) - \nabla^2 U(\theta_k)\right)(Q_w(\theta_k) - \theta_k)\right]\right\| \\
&\le \mathbf{E}\left[\left\|\left(\nabla^2 U(\theta_k + a(Q_w(\theta_k) - \theta_k)) - \nabla^2 U(\theta_k)\right)(Q_w(\theta_k) - \theta_k)\right\|\right] \\
&\le \mathbf{E}\left[\left\|\nabla^2 U(\theta_k + a(Q_w(\theta_k) - \theta_k)) - \nabla^2 U(\theta_k)\right\|_2 \|Q_w(\theta_k) - \theta_k)\|\right] \\
&\le \mathbf{E}\left[\Psi \|a(Q_w(\theta_k) - \theta_k)\| \|Q_w(\theta_k) - \theta_k)\|\right] \\
&\le \Psi \mathbf{E}\left[\|Q_w(\theta_k) - \theta_k\|^2\right] \\
&\le \frac{\Psi \Delta_w^2 d}{4}
\end{aligned}
$$

This combined with the previous result gives us

$$\|\mathbf{E}[\zeta_k]\| = \min\left(\frac{\Psi \Delta_w^2 \sqrt{d}}{4}, \frac{M \Delta_w}{2}\right).$$

Now considering the variance of $\zeta_k$,

$$
\begin{aligned}
\mathbf{E}&\left[\|\zeta_k - \mathbf{E}[\zeta_k]\|_2^2\right] \\
&\le \mathbf{E}\left[\|\zeta_k\|_2^2\right] \\
&\le \mathbf{E}\left[\left\|Q_g(\nabla \tilde{U}(Q_w(\theta_k))) - \nabla \tilde{U}(Q_w(\theta_k))\right\|_2^2\right] \\
&\quad + \mathbf{E}\left[\left\|\nabla \tilde{U}(Q_w(\theta_k)) - \nabla U(Q_w(\theta_k))\right\|_2^2\right] + \mathbf{E}\left[\|\nabla U(Q_w(\theta_k)) - \nabla U(\theta_k)\|_2^2\right] \\
&\le \frac{\Delta_g^2 d}{4} + \kappa^2 + M^2 \cdot \frac{\Delta_w^2 d}{4}.
\end{aligned}
$$

We set $\delta$ and $\sigma$ in Theorem 4 in Dalalyan & Karagulyan (2019) as the following

$$\delta = \min\left(\frac{\Psi \Delta_w^2 \sqrt{d}}{4}, \frac{M \Delta_w}{2}\right), \quad \sigma^2 d = \frac{\Delta_g^2 d}{4} + \kappa^2 + M^2 \cdot \frac{\Delta_w^2 d}{4} = \frac{(\Delta_g^2 + M^2 \Delta_w^2)d + 4\kappa^2}{4}.$$

The assumptions in Theorem 4 in Dalalyan & Karagulyan (2019) are satisfied, thus we can apply the result in Theorem 4 and get

$$W_2(\mu_K, \pi) \leq (1 - \alpha m)^K W_2(\mu_0, \pi) + 1.65(M/m)(\alpha d)^{1/2} + \frac{\delta\sqrt{d}}{m} + \frac{\sigma^2(\alpha d)^{1/2}}{1.65M + \sigma\sqrt{m}}$$

$$\leq (1 - \alpha m)^K W_2(\mu_0, \pi) + 1.65(M/m)(\alpha d)^{1/2} + \frac{\delta\sqrt{d}}{m} + \sqrt{\frac{\sigma^2 \alpha d}{m}}$$

$$= (1 - \alpha m)^K W_2(\mu_0, \pi) + 1.65(M/m)(\alpha d)^{1/2} + \min\left(\frac{\Psi\Delta_w^2 d}{4m}, \frac{M\Delta_w\sqrt{d}}{2m}\right)$$

$$+ \sqrt{\frac{(\Delta_g^2 + M^2\Delta_w^2)\alpha d + 4\alpha\kappa^2}{4m}}.$$

□

## B PROOF OF THEOREM 2

*Proof.* The update of SGLDLP-L is

$$\theta_{k+1} = Q_w\left(\theta_k - \alpha Q_g(\nabla\tilde{U}(\theta_k)) + \sqrt{2\alpha}\xi_{k+1}\right).$$

Let $\psi_{k+1} = \theta_k - \alpha Q_g(\nabla\tilde{U}(\theta_k)) + \sqrt{2\alpha}\xi_{k+1}$ so that $\theta_k = Q_w(\psi_k)$. Then,

$$\psi_{k+1} = \theta_k - \alpha Q_g(\nabla\tilde{U}(\theta_k)) + \sqrt{2\alpha}\xi_{k+1}$$
$$= Q_w(\psi_k) - \alpha Q_g(\nabla\tilde{U}(Q_w(\psi_k))) + \sqrt{2\alpha}\xi_{k+1}$$
$$= \psi_k - \alpha(\nabla U(\psi_k) + \zeta_k) + \sqrt{2\alpha}\xi_{k+1}$$

where

$$\zeta_k = \frac{\psi_k - \theta_k}{\alpha} + Q_g(\nabla\tilde{U}(\theta_k)) - \nabla U(\psi_k)$$
$$= \frac{\psi_k - \theta_k}{\alpha} + Q_g(\nabla\tilde{U}(\theta_k)) - \nabla\tilde{U}(\theta_k) + \nabla\tilde{U}(\theta_k) - \nabla U(\theta_k) + \nabla U(\theta_k) - \nabla U(\psi_k)$$

Similar to the previous proof, we know that

$$\mathbf{E}[\zeta_k] = \mathbf{E}\left[\nabla U(\theta_k) - \nabla U(\psi_k)\right] = \mathbf{E}\left[\nabla U(Q_w(\psi_k)) - \nabla U(\psi_k)\right],$$

so

$$\|\mathbf{E}[\zeta_k]\| \leq \min\left(\frac{\Psi\Delta_w^2 d}{4}, \frac{M\Delta_w\sqrt{d}}{2}\right),$$

and it suffices to set $\delta$ as above. On the other hand, the variance will be bounded by

$$\mathbf{E}\left[\|\zeta_k - \mathbf{E}[\zeta_k]\|_2^2\right] \leq \mathbf{E}\left[\left\|Q_g(\nabla\tilde{U}(\theta_k)) - \nabla\tilde{U}(\theta_k)\right\|^2\right] + \mathbf{E}\left[\left\|\nabla\tilde{U}(\theta_k) - \nabla U(\theta_k)\right\|^2\right]$$

$$+ \mathbf{E}\left[\left\|\frac{\psi_k - \theta_k}{\alpha} + \nabla U(\theta_k) - \nabla U(\psi_k)\right\|^2\right]$$

$$\leq \frac{\Delta_g^2 d}{4} + \kappa^2 + \mathbf{E}\left[\|\nabla F(\psi_k) - \nabla F(\theta_k)\|^2\right],$$

where $F(\theta) = \frac{1}{2\alpha}\|\theta\|^2 - U(\theta)$. Observe that since $U$ is $m$-strongly convex and $M$-smooth, and $\alpha^{-1} \geq M/2$, $F$ must be $h^{-1}$-smooth, and so

$$\mathbf{E}\left[\|\zeta_k - \mathbf{E}[\zeta_k]\|_2^2\right] \leq \frac{\Delta_g^2 d}{4} + \kappa^2 + \frac{1}{\alpha^2}\mathbf{E}\left[\|\psi_k - \theta_k\|^2\right]$$

$$\leq \frac{\Delta_g^2 d}{4} + \kappa^2 + \frac{\Delta_w^2 d}{4\alpha^2}.$$

This is essentially replacing the $M^2$ in the previous analysis with $\alpha^{-2}$, so, supposing the distribution of $\psi_{K+1}$ is $\nu_K$, it will give us the rate of

$$W_2(\nu_K, \pi) \leq (1 - \alpha m)^K W_2(\nu_0, \pi) + 1.65(M/m)(\alpha d)^{1/2} + \min\left(\frac{\Psi \Delta_w^2 d}{4m}, \frac{M \Delta_w \sqrt{d}}{2m}\right)$$

$$+ \sqrt{\frac{(\alpha \Delta_g^2 + \alpha^{-1} \Delta_w^2) d + 4\alpha \kappa^2}{4m}}.$$

We also have

$$W_2(\mu_k, \nu_k) = \left(\inf_{J \in \mathcal{J}(x,y)} \int \|x - y\|^2 \, dJ(x,y)\right)^{1/2} \leq \mathbf{E}\left[\|\theta_{k+1} - \psi_{k+1}\|^2\right]^{\frac{1}{2}} \leq \frac{\Delta_w \sqrt{d}}{2}.$$

Combining these two results, we get

$$W_2(\mu_K, \pi) \leq W_2(\mu_K, \nu_K) + W_2(\nu_K, \pi)$$

$$\leq (1 - \alpha m)^K W_2(\nu_0, \pi) + 1.65(M/m)(\alpha d)^{1/2} + \min\left(\frac{\Psi \Delta_w^2 d}{4m}, \frac{M \Delta_w \sqrt{d}}{2m}\right)$$

$$+ \sqrt{\frac{(\alpha \Delta_g^2 + \alpha^{-1} \Delta_w^2) d + 4\alpha \kappa^2}{4m}} + \frac{\Delta_w \sqrt{d}}{2}$$

$$\leq (1 - \alpha m)^K W_2(\mu_0, \pi) + 1.65(M/m)(\alpha d)^{1/2} + \min\left(\frac{\Psi \Delta_w^2 d}{4m}, \frac{M \Delta_w \sqrt{d}}{2m}\right)$$

$$+ \sqrt{\frac{(\alpha \Delta_g^2 + \alpha^{-1} \Delta_w^2) d + 4\alpha \kappa^2}{4m}} + ((1 - \alpha m)^K + 1) \frac{\Delta_w \sqrt{d}}{2}.$$

$\square$

## C    PROOF OF THEOREM 3

*Proof.* The update of VC SGLDLP-L is

$$\theta_{k+1} = Q^{vc}\left(\theta_k - \alpha Q_g(\nabla \tilde{U}(\theta_k)), 2\alpha, \Delta_w, Q^d, Q^s\right)$$

We ignore the variance of $Q_g$ since it is relatively small compared to weight quantization variance in practice. $Q^{vc}$ is defined as in Algorithm 1 and we have $\mathbb{E}[\theta_{k+1}] = \theta_k - \alpha \nabla U(\theta_k)$.

Let $\psi_{k+1} = \theta_k - \alpha Q_g(\nabla \tilde{U}(\theta_k)) + \sqrt{2\alpha} \xi_{k+1}$ then it follows that

$$\psi_{k+1} - \theta_{k+1} = \theta_k - \alpha Q_g(\nabla \tilde{U}(\theta_k)) + \sqrt{2\alpha} \xi_{k+1} - \theta_{k+1},$$

and

$$\psi_{k+1} = \psi_k - \alpha(\nabla U(\psi_k) + \zeta_k) + \sqrt{2\alpha} \xi_{k+1}$$

where

$$\zeta_k = \frac{\psi_k - \theta_k}{\alpha} + Q_g(\nabla \tilde{U}(\theta_k)) - \nabla U(\psi_k)$$

$$= \frac{\psi_k - \theta_k}{\alpha} + Q_g(\nabla \tilde{U}(\theta_k)) - \nabla \tilde{U}(\theta_k) + \nabla \tilde{U}(\theta_k) - \nabla U(\theta_k) + \nabla U(\theta_k) - \nabla U(\psi_k).$$

Note that $\mathbf{E}[\psi_k - \theta_k] = 0$. Similar to previous proof, we know that

$$\|\mathbf{E}[\zeta_k]\|^2 = \|\mathbf{E}[\nabla U(\theta_k) - \nabla U(\psi_k)]\|^2 \leq M^2 \mathbf{E}\left[\|\psi_k - \theta_k\|_2^2\right].$$

When $2\alpha > v_0 = \frac{\Delta_w^2}{4}$, we have that

$\mathbf{E}[\|\psi_k - \theta_k\|^2]$

$$= \mathbf{E}\left[\left\|\left(\theta_{k-1} - \alpha Q_g(\nabla \tilde{U}(\theta_{k-1}))\right) + \sqrt{2\alpha} \xi_k - Q^d\left(\theta_{k-1} - \alpha Q_g(\nabla \tilde{U}(\theta_{k-1})) + \sqrt{2\alpha - v_0} \xi_k\right) - \text{sign}(r)c\right\|^2\right]$$

Let

$$
b = Q^d \left( \theta_{k-1} - \alpha Q_g(\nabla \tilde{U}(\theta_{k-1})) + \sqrt{2\alpha - v_0}\xi_k \right)
$$
$$
- \left( \theta_{k-1} - \alpha Q_g(\nabla \tilde{U}(\theta_{k-1})) + \sqrt{2\alpha - v_0}\xi_k \right),
$$

then $|b| \leq \frac{\Delta_w}{2}$ and

$$
\mathbf{E}\left[ \|\psi_k - \theta_k\|^2 \right]
$$
$$
= \mathbf{E}\left[ \left\| \left( \theta_{k-1} - \alpha Q_g(\nabla \tilde{U}(\theta_{k-1})) \right) + \sqrt{2\alpha}\xi_k - \left( \theta_{k-1} - \alpha Q_g(\nabla \tilde{U}(\theta_{k-1})) + \sqrt{2\alpha - v_0}\xi_k \right) - b - \mathrm{sign}(r)c \right\|^2 \right]
$$
$$
= \mathbf{E}\left[ \left\| \sqrt{2\alpha}\xi_k - \sqrt{2\alpha - v_0}\xi_k - b - \mathrm{sign}(r)c \right\|^2 \right]
$$
$$
\leq \mathbf{E}\left[ \left\| \sqrt{2\alpha}\xi_k - \sqrt{2\alpha - v_0}\xi_k - b \right\|^2 \right] + \mathbf{E}\left[ \|\mathrm{sign}(r)c\|^2 \right]
$$
$$
\leq \mathbf{E}\left[ \left\| \left| \sqrt{2\alpha}\xi_k - \sqrt{2\alpha - v_0}\xi_k \right| + \frac{\Delta_w}{2} \right\|^2 \right] + v_0 d
$$
$$
\leq (\sqrt{2\alpha} - \sqrt{2\alpha - v_0})^2 \mathbf{E}[\|\xi_k\|^2] + (\sqrt{2\alpha} - \sqrt{2\alpha - v_0})\Delta_w \mathbf{E}[|\xi_k|] + 2v_0 d
$$
$$
\leq \left( (\sqrt{2\alpha} - \sqrt{2\alpha - v_0})^2 + (\sqrt{2\alpha} - \sqrt{2\alpha - v_0})\Delta_w + 2v_0 \right) d
$$

Since $2xy \leq x^2 + y^2$, we get

$$
(\sqrt{2\alpha} - \sqrt{2\alpha - v_0})\Delta_w \leq (\sqrt{2\alpha} - \sqrt{2\alpha - v_0})^2 + \frac{\Delta_w^2}{4} = (\sqrt{2\alpha} - \sqrt{2\alpha - v_0})^2 + v_0
$$

It follows

$$
\mathbf{E}[\|\psi_k - \theta_k\|^2] \leq \left( 2(\sqrt{2\alpha} - \sqrt{2\alpha - v_0})^2 + 3v_0 \right) d.
$$

Note that

$$
\sqrt{2\alpha} - \sqrt{2\alpha - v_0} = \frac{2\alpha - (2\alpha - v_0)}{\sqrt{2\alpha} + \sqrt{2\alpha - v_0}} = \frac{v_0}{\sqrt{2\alpha} + \sqrt{2\alpha - v_0}} \leq \frac{v_0}{\sqrt{2\alpha}}.
$$

The expectation becomes

$$
\mathbf{E}[\|\psi_k - \theta_k\|^2] \leq \left( \frac{v_0^2}{\alpha} + 3v_0 \right) d.
$$

Since $2\alpha > v_0$, it follows

$$
\mathbf{E}[\|\psi_k - \theta_k\|^2] \leq (2v_0 + 3v_0) d = 5v_0 d.
$$

Let $A = 5v_0 d$. Then

$$
\|\mathbf{E}[\zeta_k]\|^2 \leq M^2 \cdot A,
$$

and also

$$
\|\mathbf{E}[\zeta_k]\| \leq \Psi \cdot A.
$$

Therefore,

$$
\delta = \min \left( \Psi A, M\sqrt{A} \right).
$$

The variance will be bounded by

$$
\mathbf{E}\left[ \|\zeta_k - \mathbf{E}[\zeta_k]\|_2^2 \right] \leq \mathbf{E}\left[ \left\| Q_g(\nabla \tilde{U}(\theta_k)) - \nabla \tilde{U}(\theta_k) \right\|^2 \right] + \mathbf{E}\left[ \left\| \nabla \tilde{U}(\theta_k) - \nabla U(\theta_k) \right\|^2 \right]
$$
$$
+ \mathbf{E}\left[ \left\| \frac{\psi_k - \theta_k}{\alpha} + \nabla U(\theta_k) - \nabla U(\psi_k) \right\|^2 \right]
$$
$$
\leq \frac{\Delta_g^2 d}{4} + \kappa^2 + \frac{1}{\alpha^2} \mathbf{E}\left[ \|\psi_k - \theta_k\|^2 \right]
$$
$$
\leq \frac{\Delta_g^2 d}{4} + \kappa^2 + \frac{A}{\alpha^2}.
$$

Supposing the distribution of $\psi_{K+1}$ is $\nu_K$, it will give us the rate of

$$W_2(\nu_K, \pi) \le (1 - \alpha m)^K W_2(\nu_0, \pi) + 1.65(M/m)(\alpha d)^{1/2} + \min\left(\frac{\Psi \cdot A}{m}, \frac{M\sqrt{A}}{m}\right)$$

$$+ \sqrt{\frac{\alpha \Delta_g^2 d + 4\alpha\kappa^2}{4m} + \frac{A}{\alpha m}}.$$

We also have

$$W_2(\mu_K, \nu_K) = \left(\inf_{J \in \mathcal{J}(x,y)} \int \|x - y\|^2 \, dJ(x,y)\right)^{1/2} \le \mathbf{E}\left[\|\theta_{K+1} - \psi_{K+1}\|^2\right]^{\frac{1}{2}} \le \sqrt{A}.$$

Combining these two results, we get

$$W_2(\mu_K, \pi) \le W_2(\mu_K, \nu_K) + W_2(\nu_K, \pi)$$

$$\le (1 - \alpha m)^K W_2(\nu_0, \pi) + 1.65(M/m)(\alpha d)^{1/2} + \min\left(\frac{\Psi \cdot A}{m}, \frac{M\sqrt{A}}{m}\right)$$

$$+ \sqrt{\frac{\alpha \Delta_g^2 d + 4\alpha\kappa^2}{4m} + \frac{A}{\alpha m}} + \sqrt{A}$$

$$\le (1 - \alpha m)^K W_2(\mu_0, \pi) + 1.65(M/m)(\alpha d)^{1/2} + \min\left(\frac{\Psi \cdot A}{m}, \frac{M\sqrt{A}}{m}\right)$$

$$+ \sqrt{\frac{\alpha \Delta_g^2 d + 4\alpha\kappa^2}{4m} + \frac{A}{\alpha m}} + \left((1 - \alpha m)^K + 1\right)\sqrt{A}.$$

When $2\alpha < \frac{\Delta_w^2}{4}$, since we assume that the gradient is bounded by $\mathbf{E}\left[\left\|Q_g(\nabla\tilde{U}(\theta_k))\right\|_1\right] \le G$

$$\mathbf{E}[\|\psi_k - \theta_k\|^2] = \mathbf{E}\left[\left\|\left(\theta_{k-1} - \alpha Q_g(\nabla\tilde{U}(\theta_{k-1}))\right) - \theta_k + \sqrt{2\alpha}\xi_k\right\|^2\right]$$

$$= \mathbf{E}\left[\left\|\left(\theta_{k-1} - \alpha Q_g(\nabla\tilde{U}(\theta_{k-1}))\right) - \theta_k\right\|^2\right] + \mathbf{E}\left[\left\|\sqrt{2\alpha}\xi_k\right\|^2\right]$$

$$\le \max\left(2\mathbf{E}\left[\left\|\left(\theta_{k-1} - \alpha Q_g(\nabla\tilde{U}(\theta_{k-1}))\right) - Q^s\left(\theta_{k-1} - \alpha Q_g(\nabla\tilde{U}(\theta_{k-1}))\right)\right\|^2\right], 4\alpha d\right)$$

Using the bound equation (6) in Li & De Sa (2019),

$$\mathbf{E}\left[\left\|\left(\theta_{k-1} - \alpha Q_g(\nabla\tilde{U}(\theta_{k-1}))\right) - Q^s\left(\theta_{k-1} - \alpha Q_g(\nabla\tilde{U}(\theta_{k-1}))\right)\right\|^2\right]$$

$$\le \Delta_w \alpha \mathbf{E}\left[\left\|Q_g(\nabla\tilde{U}(\theta_{k-1}))\right\|_1\right]$$

$$\le \Delta_w \alpha G$$

It follows

$$\mathbf{E}[\|\psi_k - \theta_k\|^2] \le \max\left(2\Delta_w \alpha G, 4\alpha d\right)$$

Let $A = \max\left(2\Delta_w \alpha G, 4\alpha d\right)$. The rest is that same as the case $2\alpha > v_0$.

$\square$

# D    COMPARISON TO SGD BOUNDS

Following previous work in comparing sampling and optimization methods (Ma et al., 2019; Talwar, 2019), we also compare our 2-Wasserstein distance bound with previosu SGD bounds. Previous low-precision SGD convergence bounds are shown in terms of the squared distance to the optimum $\left\|\bar{\theta}_K - \theta^*\right\|_2^2$ (Yang et al., 2019). In order to compare our bounds with theirs, we consider a 2-Wasserstein distance between two point distributions. Let $\mu_K$ be the point distribution assigns zero

probability everywhere except $\bar{\theta}_K$ and $\pi$ be the point distribution assigns zero probability everywhere except $\theta^*$. Then we get

$$W_2(\mu_K, \pi) = \left( \inf_{J \in \mathcal{J}(x,y)} \int \|x - y\|^2 \, dJ(x,y) \right)^{1/2} \leq \mathbf{E}\left[ \left\| \bar{\theta}_K - \theta^* \right\|^2 \right]^{\frac{1}{2}}.$$

From (Yang et al., 2019), we know that $\mathbf{E}\left[ \left\| \bar{\theta}_K - \theta^* \right\|^2 \right]^{\frac{1}{2}}$ is proportional to $\Delta_w$. Therefore, our 2-Wasserstein distance is $O(\Delta_w^2)$ whereas SGD's 2-Wasserstein distance is $O(\Delta_w)$, which shows SGLD is more robust to the quantization error.

## E  ALGORITHMS WITH (BLOCK) FLOATING POINT NUMBERS

The qunatization gap is floating point and block floating point is computed as

$$\Delta_w(\mu) \leftarrow \begin{cases} 2^{E[\mu]-W+2} \text{ where } E[\mu] = \mathrm{clip}(\lfloor \log_2(\max |\mu|) \rfloor, l, u) & \text{block floating point} \\ 2^{E[\mu]-W} \text{ where } E[\mu] = \mathrm{clip}(\lfloor \log_2(|\mu|) \rfloor, l, u) & \text{floating point} \end{cases} \tag{3}$$

Then deterministic rounding and stochastic rounding are defined with $\Delta_w$. VC SGLDLP-L with (block) floating point is outlined in Algorithm 3

---

**Algorithm 3** VC SGLDLP-L with (Block) Floating Point.

**given:** Stepsize $\alpha$, number of training iterations $K$, gradient quantizer $Q_g$, deterministic rounding with (block) floating point $Q^d$, stochastic rounding with (block) floating point $Q^s$, $F$ bits to represent the shared exponent (block floating point) or the exponent (floating point), $W$ bits to represent each number in the block (block floating point) or the mantissa (floating point).
**let** $l \leftarrow -2^{F-1}, u \leftarrow 2^{F-1} - 1$
**for** $k = 1 : K$ **do**
    **compute** $\mu \leftarrow \theta_k - \alpha Q_g \left( \nabla \tilde{U}(\theta_{k-1}) \right)$
    **compute** $\Delta_w(\mu)$ following Equation (3)
    **update** $\theta_{k+1} \leftarrow Q^{vc}\left( \mu, 2\alpha T, \Delta_w(\mu), Q^d, Q^s \right)$
**end for**
**output:** samples $\{\theta_k\}$

---

**Algorithm 4** Variance-Corrected Quantization Function $Q^{vc}$ with (Block) Floating Point.

**input:** $(\mu, v, \Delta, Q^d, Q^s)$
$v_0 \leftarrow \Delta^2/4$
**if** $v > v_0$ **then**
    $x \leftarrow \mu + \sqrt{v - v_0}\xi$, where $\xi \sim \mathcal{N}(0, I_d)$
    $r \leftarrow x - Q^d(x)$
    recompute $\Delta \leftarrow \Delta_w(x)$ following Equation (3)
    $c \leftarrow \begin{cases} \Delta, & w.p. \frac{v_0 + r^2 + |r|\Delta}{2\Delta^2} \\ -\Delta, & w.p. \frac{v_0 + r^2 - |r|\Delta}{2\Delta^2} \\ 0, & \text{otherwise} \end{cases}$
    **return** $Q^d(x) + \mathrm{sign}(r) \cdot c$
**else**
    the same as in fixed point numbers
**end if**

---

The $Q^{vc}$ function with (block) floating point in Algorithm 4 is the same as Algorithm 1 except the line in red. We recompute the quantization gap $\Delta$ after adding Gaussian noise to make sure it aligns with the quantization gap of $x$.

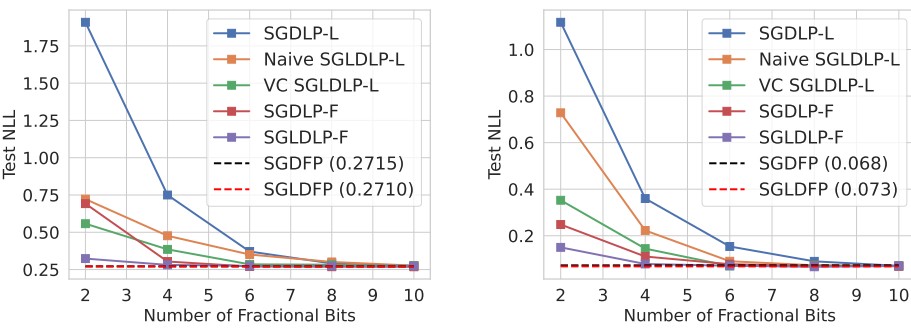

Figure 3: Test NLL on MNIST dataset in terms of different precision.

# F    EXPERIMENTAL DETAILS AND ADDITIONAL RESULTS

## F.1    SAMPLING METHODS

For both SGLD and low-precision SGLD, we collected samples $\{\theta_k\}_{K=1}^J$ from the posterior of the model's weight, and obtained the prediction on test data $\{x^*, y^*\}$ by Bayesian model averaging

$$p(y^*|x^*, \mathcal{D}) \approx \frac{1}{J} \sum_{j=1}^{J} p(y^*|x^*, \mathcal{D}, \theta_j).$$

## F.2    MNIST

We train all methods on logistic regression and MLP for 20 epochs with learning rate 0.1 and batch size 64. We additionally report negative log-likelihood (NLL) comparisons in Figure 3.

## F.3    CIFAR AND IMDB

For CIFAR datasets, we use batch size 128, learning rate $0.5$ and weight decay $5e - 4$. We train the model for 245 epochs and used the same decay learning rate schedule as in Yang et al. (2019). We collect 50 samples for SGLD. For cyclical learning rate schedule, we use 7 cycles and collect 5 models per cycle (35 models in total).

For IMDB dataset, we use batch size 80, learning rate $0.3$ and weight decay $5e - 4$. We use a two-layer LSTM. The embedding dimension and the hidden dimension are 100 and 256 respectively. We train the model for 50 epochs and used the same decay learning rate schedule as on CIFAR datasets. We collect 20 samples for SGLD. For cyclical learning rate schedule, we use 1 cycles and collect 20 models.

## F.4    IMAGENET

We use batch size 256, learning rate $0.2$ and weight decay $1e - 4$. We use the same decay learning rate schedule as in He et al. (2016) and collect 20 models for SGLD.

