# OpenReview forum: "Low-Precision Stochastic Gradient Langevin Dynamics"
_ICLR.cc/2022/Conference — ICLR 2022 Submitted_

### Official Review · Reviewer_dB5g · 2021-10-25

**Correctness:** 3
**Technical Novelty And Significance:** 2
**Empirical Novelty And Significance:** 2
**Recommendation:** 3
**Confidence:** 4

**Main Review:**



Pros: the authors conducted some bias analysis for low-precision SGLD based on [Dalalyan and Karagulyan 19]'s result.

Cons:

1. **Limited novelty in methodology**: the novelty in terms of methodology is limited and is not very interesting. Simply extending quantization results from SGD to SGLD with further indications is not that interesting.

2. **Limited depth in theory**: the theoretical novelty is limited and has limited depth; although I am very familiar with Dalalyan's work, the authors don't even bother to write down Theorem 4 and the relevant background in  [Dalalyan and Karagulyan 19].

3. **Inconsistencies between experiments and theories**: The experiments are mainly running in non-convex settings, which don't match the theoretical claim in strongly convex scenarios. If I were the authors, I would only run MNIST experiments based on logistic regression; unless I have theoretical results that extend to non-convex settings. Because the theory only claims these contributions in strongly-convex settings.

4. **Incomplete experiments**: Running SGLD for reproducing purely optimization results seems to be not interesting and uncertainty estimations would also be appreciated.

**Summary Of The Paper:**

The author proposed a comprehensive study of low-precision Stochastic Gradient Langevin Dynamics (SGLD). They proposed the convergence analysis of low-precision Stochastic Gradient Langevin Dynamics on strongly log-concave distributions.  Motivated by the diverging fact of SGLDLP-L with small stepsizes, they proposed a variance-corrected quantization function to ensure a bounded bias. Empirical experiments based on large-scale DNN examples are evaluated.

**Summary Of The Review:**

The paper proposed some analysis for low-precision SGLD but the novelties or depths in terms of methodology and theory are both limited. In addition, the experiments don't match the theoretical analysis and don't have the results for uncertainty estimations, which leads me to suspect why to bother to use SGLD (I believe SGD with fine-tuning parameters can also achieve the claimed results). Based on these evaluations, I tend to reject this paper.

---

> ### Author Response · Authors · 2021-11-23
> **Response to Reviewer dB5g**
>
> Thank you for your insightful feedback. We answer your questions below.
>
> Q1: Limited novelty in methodology.
>
> A1: We would like to emphasize that it is non-trivial to speed up SGLD by low-precision while guaranteeing convergence to the target distribution. As we show in Section 4.3, naively applying quantization methods in optimization to SGLD can result in arbitrary divergence from the target distribution. We identify the source of the issue, and propose a new quantization function to correct the bias of low-precision SGLD. Moreover, our solution is simple to implement and is cheap to use in deep learning. For the first time, our method enabled fully quantized SGLD, achieving comparable results with full-precision SGLD and outperforming low-precision SGD on several Bayesian deep learning benchmarks.
>
>
> We believe that the methodology in the paper is sufficiently novel, which provides insights on the key issue of low-precision sampling, and a practical approach that can be cheaply used on large-scale deep learning tasks.
>
>
>
>
> Q2: Limited depth in theory.
>
> A2: To the best of our knowledge, our theoretical results are the first results showing how low-precision computation affects the convergence of sampling methods. The meaning of our theoretical results are three-fold: (1) it ensures that SGLD with full-precision gradient accumulators is convergent and can be safely used in practice; (2) it reveals the problem of naively using low-precision gradient accumulators in SGLD. Inspired by it, we propose a new theoretically guaranteed solution to enable fully quantized SGLD; (3) it shows the dependency of convergence on the quantization gap, indicating that SGLD is more robust than SGD when using full-precision gradient accumulators. We believe that these theoretical results are important for understanding low-precision sampling and providing guidance about its use in practice.
>
> We have added more background information of  [Dalalyan & Karagulyan, 2019] in the revision.
>
>
> Q3: Inconsistencies between experiments and theories
>
>
> A3: We have conducted experiments on a Gaussian distribution and a logistic regression to empirically verify the theoretical results. Since one of the main use cases of low-precision is to speed up deep learning, we also tested the proposed methods on standard Bayesian deep learning tasks. We believe the results in deep learning will be of interest to practitioners. Please note that though low-precision optimization has been widely used in practice, as far as we understand, there is no theoretical convergence result of low-precision optimization in a non-convex setting. We agree that getting theoretical results of the proposed methods in a non-convex setting is very interesting and important, and we hope our promising empirical results would motivate more study on this line.
>
> Q4: Incomplete experiments
>
> A4: Our experiments contain (1) a synthetic distribution, which verifies that our methods can converge to the target distribution; (2) a logistic regression, which verifies our theoretical results; (3) Bayesian neural networks on image and text classification, which demonstrate the effectiveness of our methods on Bayesian deep learning benchmarks.
>
> We believe that getting classification results with BNNs is interesting since it tests the generalization of BNNs on popular deep learning tasks. In fact, image classification is one of the most common benchmarks in Bayesian deep learning and has been included in most previous papers on SGMCMC in Bayesian deep learning [1,2,3]. Our empirical results show that low-precision SGLD largely preserves the generalization performance of full-precision SGLD and is more robust to quantization error than optimization.
>
> We agree that uncertainty estimation is also interesting and will add an additional experiment on it in the revision.
>
> [1] Preconditioned Stochastic Gradient Langevin Dynamics for Deep Neural Networks. AAAI 2016
>
> [2] Bayesian Inference for Large Scale Image Classification. 2019
>
> [3] Cyclical Stochastic Gradient MCMC for Bayesian Deep Learning. ICLR 2020

---

### Official Review · Reviewer_yqrt · 2021-10-29

**Correctness:** 3
**Technical Novelty And Significance:** 2
**Empirical Novelty And Significance:** 2
**Recommendation:** 3
**Confidence:** 4

**Main Review:**

Strengths:
- The narrative is fluent and flow of statement is smooth and natural. In general, the writing of the manuscript is well done.
- The authors have tried their proposed algorithms in multiple classic application scenarios.

Weaknesses:
- The submission does not appear to be a major contribution to the field, as the main novel point is the analysis of the variance of low-precision gradient estimator.

Specifics:
- page 3, section 3.2, paragraph 3, line 3: $l = -2^{W-F-2}$, why the power is $-2$ instead of $-1$ when computing the upper bound of presentable number?

- page 5, section 4.2, the line below the statement of theorem 2: I think theorem 2 does not suggest "SGLDLP-L diverges from the target distribution", but instead, it shows the bound becomes loose when stepsize $\alpha$ decreases.

- page 4 and 5: schemes SGLDLP-F shown in equation (1) and SGLDLP-L shown in equation (2) essentially only differs in whether the noise $\xi_{k}$ is low-precision thresholded, doesn't it?

- The main theorems are proved with the results from Dalalyan and Karagulyan's work. The author should explicitly cite the specific theorem in [DK19] and explain why the argued variance of low-precision gradient estimator renders the conditions of the cited theorem satisfied and thus the theorem applicable.

**Summary Of The Paper:**

This paper discusses using the stochastic gradient Langevin dynamics with low-precision implementation. The authors present convergence results in Wasserstein distance norm for three implementation cases, which distinguish specific low-precision thresholding schemes and different quantization techniques. The authors also report simulation outcome of proposed low-precision SGLD algorithms and show their convergence in actual application.

**Summary Of The Review:**

In light of the specifics in the main review, the discussion of low-precision SGLD in this submission is not significant enough and there are some places in the argument that are not clear.

---

> ### Author Response · Authors · 2021-11-23
> **Response to Reviewer yqrt**
>
> Thank you for your valuable review. We answer your questions below.
>
> Q1: The submission does not appear to be a major contribution to the field.
>
> A1: We would like to emphasize the significance and novelty of our work. While low-precision optimization has developed rapidly, sampling, as another popular inference tool, remains largely unexplored with low-precision computation. To the best of our knowledge, this paper is the first comprehensive investigation for low-precision stochastic gradient MCMC, considering both theoretical convergence and practical performance in deep learning. Our contributions are three-fold: (1) we provide a methodology for SGLD to leverage low-precision computation while still guaranteeing the convergence to the target distribution; (2) we offer theoretical results which explicitly show how quantization gap affects the convergence of the sampler; (3) we conducted multiple experiments including a synthetic distribution, a logistic regression, and Bayesian neural networks on image and text classification. These results for the first time empirically demonstrate the effectiveness of low-precision SGLD and suggest that it is particularly suitable with low-precision deep learning because of its natural ability to handle system noise.
>
> Therefore, we argue that this paper makes a sufficient contribution and represents a step towards fast and efficient sampling in Bayesian deep learning.
>
>
> Q2: Why the power is  -2 instead of  -1 when computing the upper bound of presentable number?
>
> A2: Thanks for pointing it out. This is a typo and it should be -1 because we use one bit to represent the sign. We have corrected it.
>
> Q3: I think theorem 2 does not suggest "SGLDLP-L diverges from the target distribution", but instead, it shows the bound becomes loose when stepsize decreases.
>
> A3: We agree that this is one possible explanation for the bound in Theorem 2. However, our empirical results in Figure 1 suggest that SGLDLP-L indeed diverges from the target distribution, and the divergence increases as the stepsize decreases. This verifies that the distance increases due to the divergence from the target distribution rather than the loose bound. We have clarified this point in the paper.
>
>
> Q4: SGLDLP-F shown in equation (1) and SGLDLP-L shown in equation (2) essentially only differ in whether the noise is low-precision thresholded.
>
> A4: No, SGLDLP-F and  SGLDLP-L do not only differ in whether the noise is in low-precision. SGLDLP-F uses a full-precision weight buffer to accumulate each update and only quantizes the weight before computing the gradient while SGLDLP-L keeps the weight in low-precision discrete space.
>
> Specifically, even if the noise $\sqrt{2\alpha}\xi_{k+1}$ is in low-precision, in SGLDLP-F, $\theta_{k+1} = \theta_{k} - \alpha Q_g\left(\nabla\tilde{U}(Q_w\left(\theta_{k})\right)\right) + \sqrt{2\alpha}\xi_{k+1}$ is not in low-precision since  $\theta_k$ and $\alpha Q_g\left(\nabla\tilde{U}\right)$ are not in low-precision. On the other hand, $\theta_{k+1}$ in SGLDLP-L is always represented in low-precision.
>
>
> Q5: The author should explicitly cite the specific theorem in [DK19] and explain why the argued variance of low-precision gradient estimator renders the conditions of the cited theorem satisfied and thus the theorem applicable.
>
>
> A5: Thanks for the comments. We have improved the presentation of proofs accordingly.

---

### Official Review · Reviewer_bvZD · 2021-11-03

**Correctness:** 3
**Technical Novelty And Significance:** 4
**Empirical Novelty And Significance:** 3
**Recommendation:** 6
**Confidence:** 4

**Main Review:**

Posterior sampling of quantized neural networks is an important problem to study. While the optimization of quantized neural networks are relatively well studied, this paper is the first to consider the sampling of such networks. The technical quality of the is nice. It presents reasonable convergence results for the proposed SGLDLP algorithms. The idea of utilizing the quantization noise for sampling and variance correction makes sense.

In terms of weakness, I think the assumptions in the paper are worth considering:
1. the gradient quantizer assumes that we first have a full-precision gradient, and then quantize it, maybe for reducing the communication cost for distributed training. The formulation does not support applications such as mixed precision training, where the gradient itself is computed in an approximate way. Is distributed training the case the authors want to address?
2. While the convergence result presented in Theorem 1 is good, it relies on a non-converging term relates to Delta_w. It is curious to compare with the following post-training quantization algorithm:
a' generate full precision samples {theta_k} with full-precision SGLD
b' use the quantized version {Q(theta_k)} as the low-precision samples;
We can imagine that the posterior distribution given by this algorithm is also close enough to the true posterior, where the distance is controlled by Delta_w. It would be better if we can show the proposed SGLDLP is better than the post-training quantization algorithm.

Post rebuttal
====

Thanks for addressing my concerns. I am tending to keep my score since I think it is an interesting paper with some novelties, but the novelty is limited, as commented by other reviewers.

**Summary Of The Paper:**

This paper investigate posterior sampling of weight-quantized neural networks with SGLD. The paper gives the convergence of SGLD under Wasserstein distance with quantized weights and gradients. It also considers the situation when the accumulator is also in low precision, and proposes a variance correction strategy to deal with the additional quantization noise.

**Summary Of The Review:**

A novel paper with reasonable techniques. The results might be improved by tightening the assumptions.

---

> ### Author Response · Authors · 2021-11-23
> **Response to Reviewer bvZD**
>
> Thanks for your supportive and thoughtful comments. We answer your questions below.
>
> Q1: The gradient quantizer assumes that we first have a full-precision gradient, and then quantize it.
>
> A1: We do not assume having a full-precision gradient. The formulation in the paper follows previous low-precision training papers e.g. [1,2], where in backward propagation, the error and gradient in each layer are quantized. As shown in these previous works, doing so can significantly reduce computational and memory costs of gradient computation. We do not consider distributed learning in the paper. We have clarified this part in the revision.
>
> [1] Training and Inference with Integers in Deep Neural Networks. ICLR 2018
>
> [2] Swalp: Stochastic Weight Averaging in Low-Precision Training. ICML 2019
>
>
> Q2: Theoretical comparison with post-training quantization algorithm.
>
> A2: In this paper, we focus on training low-precision models by SGLD from scratch, which can reduce both training and testing costs. Post-training quantization is another common strategy in low-precision, but only reduces testing cost. We agree that getting theoretical results for SGLD with post-training quantization is interesting and we leave it for future work. As far as we understand, for SGD, there is also no theoretical benefit of low-precision training over post-training quantization even though the former generally performs better in deep learning.

---

### Official Review · Reviewer_gLYz · 2021-11-04

**Correctness:** 2
**Technical Novelty And Significance:** 2
**Empirical Novelty And Significance:** 2
**Recommendation:** 3
**Confidence:** 4

**Main Review:**

I think the idea of applying low-precision optimization to SGLD is a direction worth exploring, but the overall presentation of the paper seems to be vague to me. While SGLD is viewed as an optimization algorithm particularly for nonconvex optimization problem which aims to minimize an objective function $U$ (see e.g., Raginsky et al., 2017), Langevin Monte Carlo (LMC), which takes exact the same algorithmic form as SGLD, is intrinsically a sampling algorithm which aims to sample the stationary distribution $\exp(-U)$. The authors has been mixing up the tasks of optimization and sampling in the theoretical and experimental results, making it hard to understand what the task the proposed algorithm wants to solve. In the theoretical results of this paper, the authors leverage results in Dalalyan and Karagulyan (2019) to provide a (sampling) bound on the Wasserstein distance between the distribution generated by the low-precision SGLD and the target distribution, but compare this with an optimization bound in Yang et al. (2019), which is the distance between the SGD estimation and the optimum. This seems very weird to me since I think these are not comparable.

Then, in the experiments, I think the authors are applying the proposed low-precision SGLD as a training (i.e., optimization) algorithm instead of a sampling algorithm. The metric used (test error) also does not seem to measure the accuracy of the sampled distribution using low-precision SGLD. Thus, because of this ambiguity between applying SGLD as a sampling algorithm versus an optimization (training) algorithm, the current results in this work are quite questionable to me.

Some mathematical statements and notation in this paper are also unclear. In the assumption, the authors did not mention that the third condition is that the function $U$ needs to have a Lipschitz Hessian. Representation of the stochastic gradient of $U$ by $\nabla\tilde{U}$ is also weird since you actually did not change the function $U$ to $\tilde{U}$, but the gradient to a stochastic gradient. A more commonly used notation should be $\nabla\widetilde{U}$. Also $\mu_0$ is not defined in Theorem 1 (and typed as $\nu_0$ in Theorem 2 which must be a typo). The derivation in Section 4.3 must be wrong as well, since $\theta_{k+1}$ is a vector-valued random variable, its variance must be matrix-valued. Thus, the bottom part of page 5 is wrong. Also, using $\theta_{k+1}$ to represent both the updates with and without quantization have been leading to much confusion during my review.


---
Raginsky, Maxim, Alexander Rakhlin, and Matus Telgarsky. "Non-convex learning via stochastic gradient Langevin dynamics: a nonasymptotic analysis." Conference on Learning Theory. PMLR, 2017.


**Summary Of The Paper:**

This paper proposes a low-precision version of SGLD which makes use of low-bit arithmetic. In particular, since low-precision gradient accumulators might lead to divergence of SGLD, the authors propose a new quantization function to preserve the correct variance in each update step.

**Summary Of The Review:**

The idea of  applying low-precision optimization to SGLD is a direction worth exploring, but the ambiguity between applying SGLD as a sampling algorithm versus an optimization (training) algorithm in this paper has made the current results in this work quite questionable. Some mathematical statements and notation in this paper are also unclear.

---

> ### Author Response · Authors · 2021-11-23
> **Response to Reviewer gLYz**
>
> Thank you for your thoughtful review. We answer your questions below.
>
> Q1: Mixing up the tasks of optimization and sampling in the theoretical and experimental results. Comparison with an optimization bound in Yang et al. (2019).
>
> A1: We would like to clarify that we consider the sampling task in both theory and practice, where the goal is to sample from the target distribution (which is the posterior distribution in the Bayesian setup). SGLD in this paper is viewed as a sampling method.
>
> There have been many works on comparing optimization and sampling algorithms since they serve as two main computational strategies for machine learning [1,2]. For example in [1], the authors compare the total variation distance between the approximate distribution and the target distribution (sampling bound), with the objective gap $U(\theta_K) - U(\theta^*)$ (optimization bound). Our comparison between SGLD and SGD is similar, where we compare the distance between the approximate distribution and the target distribution, with the distance between SGD estimation and the optimum. Therefore, we believe our comparison is reasonable. Furthermore, we think this comparison is essential since it implies how sensitive SGLD and SGD are to the quantization error, and actually suggests that sampling methods are more suitable with low-precision computation.
>
> We have added these explanations in the revision to clarify the comparison with optimization.
>
> [1] Sampling can be faster than optimization. PNAS 2019
>
> [2] Computational Separations between Sampling and Optimization. NeurIPS 2019
>
> Q2: In the experiments, I think the authors are applying the proposed low-precision SGLD as a training (i.e., optimization) algorithm instead of a sampling algorithm.
>
> A2: We use the proposed low-precision SGLD as a sampling algorithm in all experiments. Specifically in Section 5, for both SGLD and low-precision SGLD, we collected samples from the posterior of the model’s weight, and obtained the prediction on test data by Bayesian model averaging. Since the true posterior is intractable, test error is typically used as a metric to evaluate sampling quality. Our experimental setup is standard and is consistent with previous work on SGMCMC in Bayesian machine learning [1,2,3]. Our empirical results demonstrate that the proposed low-precision SGLD is comparable to full-precision SGLD, indicating that the sampling quality is mostly maintained even with low-precision computation. It is also better than low-precision SGD, showing that sampling is more robust to the quantization error than optimization.
>
> Besides, in Figure 1, where the target distribution is known, we can clearly see that our low-precision SGLD can estimate the target distribution very accurately.
>
> We have added more experimental details of SGLD and low-precision SGLD in the revision.
>
> [1] Preconditioned Stochastic Gradient Langevin Dynamics for Deep Neural Networks. AAAI 2016
>
> [2] Bayesian Inference for Large Scale Image Classification. 2019
>
> [3] Cyclical Stochastic Gradient MCMC for Bayesian Deep Learning. ICLR 2020
>
> Q3: Some mathematical statements and notations in this paper are also unclear.
>
> A3: Thanks for the suggestions. We have revised the statements and notations in the paper. In each step, $\theta_{k+1}$ can be viewed as sampling from a Gaussian distribution with diagonal covariance matrix $2\alpha I$.  If we consider each dimension of $\theta_{k+1}$, the derivation in Section 4.3 still holds. We have revised this part to avoid confusion.

---

### Author Response · Authors · 2021-11-25
**Summary**

We thank all reviewers for their constructive review and have revised the paper accordingly. The main changes are the following:

- Section 4. We revised the mathematical statements and notations to avoid confusion.

- Appendix A. We added more background information of [Dalalyan & Karagulyan, 2019].

- Appendix D. We added more explanations on the comparison with the optimization bound.

- Appendix F. We added more information about the implementation of sampling methods on Bayesian neural networks.

Finally, we would like to clarify the contribution and the potential impact of this paper. While we appreciate the thoughtful comments, we believe key aspects of our contributions, especially the bigger picture context, have been overlooked. Low-precision optimization has developed rapidly, and has had an extraordinary impact. Sampling, by contrast, remains largely unexplored with low-precision computation. For this reason, sampling is simply infeasible in many scenarios, especially involving deep neural networks, such that practitioners have to resort to optimization.
This is all despite the fact that sampling can provide the greatest benefits to generalization performance when we are using large neural networks, and moreover is perhaps even more naturally coupled with low-precision computations than optimization.

To the best of our knowledge, this paper is the first comprehensive investigation for low-precision stochastic gradient MCMC (SGMCMC), providing both theoretical convergence bounds and promising empirical results in deep learning. We note that SGMCMC can especially benefit from low-bit arithmetic thanks to its intrinsic ability to handle system noise. We show that by doing so, the computational and memory costs of SGMCMC can be remarkably reduced without sacrificing its original performance. This will particularly enable sampling with very large architectures in deep learning which is currently infeasible. Our variance correction is nice but is only one small aspect of our general contribution, and is perhaps being perceived quite narrowly as the focus of our paper. We believe this work fills an important gap in the literature, and will significantly accelerate the practical use of sampling methods on large-scale and resource-restricted machine learning problems. We believe the simplicity of our approach is in fact a strength rather than a weakness, and we hope reviewers can be open-minded in considering the larger context of our paper in their final consideration of our work. We will try to clarify this context further in the final version. Addressing the topic of low precision stochastic MCMC, alongside clear demonstrations of what is possible, is in itself a really valuable contribution --- particularly in the context of work on low precision optimization, and the relative lack of work on low precision MCMC, despite its promise in this setting.

---

### Decision · Program_Chairs · 2022-01-20

**Decision:**

Reject

**Comment:**

The paper investigates the performance of low-precision Stochastic Gradient Langevin Dynamics (SGLD). While similar low-precision techniques have been widely used in optimization, much less is known for Markov Chain Monte Carlo (MCMC) methods. The paper develops a new quantization function to make SGLD suitable for low-precision setups and argues for its use in deep learning.

The main concerns among the reviewers were related to the paper presentation (separation and comparison between optimization and sampling), comparison to Dalalyan-Karagulyan'19 and overview of this work, technical depth, and numerical experiments. The authors have adequately responded to the reviewers' comments and addressed them to the extent possible. However, there was ultimately not enough support to lead this paper to acceptance.

I find low-precision sampling a worthy topic of study and the contributions of the paper are interesting. The authors are encouraged to revise the paper based on the reviewers' comments, more clearly highlight the contributions, and resubmit.